# LiDAR-Based Urban Three-Dimensional Rail Area Extraction for Improved Train Collision Warnings

**DOI:** 10.3390/s24154963

**Published:** 2024-07-31

**Authors:** Tuo Shen, Jinhuang Zhou, Tengfei Yuan, Yuanxiang Xie, Xuanxiong Zhang

**Affiliations:** 1School of Optical-Electrical and Computer Engineering, University of Shanghai for Science and Technology, Shanghai 200093, China; shengtuo339@tongji.edu.cn (T.S.); xxzhang@usst.edu.cn (X.Z.); 2Shanghai Key Laboratory of Rail Infrastructure Durability and System Safety, Shanghai 201804, China; 2133387@tongji.edu.cn (J.Z.); 2233390@tongji.edu.cn (Y.X.); 3SILC Business School, Shanghai University, Shanghai 201800, China

**Keywords:** LiDAR, track area, extraction method, urban rail transit

## Abstract

The intrusion of objects into track areas is a significant issue affecting the safety of urban rail transit systems. In recent years, obstacle detection technology based on LiDAR has been developed to identify potential issues, in which accurately extracting the track area is critical for segmentation and collision avoidance. However, because of the sparsity limitations inherent in LiDAR data, existing methods can only segment track regions over short distances, which are often insufficient given the speed and braking distance of urban rail trains. As such, a new approach is developed in this study to indirectly extract track areas by detecting references parallel to the rails (e.g., tunnel walls, protective walls, and sound barriers). Reference point selection and curve fitting are then applied to generate a reference curve on either side of the track. A centerline is then extrapolated from the two curves and expanded to produce a 2D track area with the given size specifications. Finally, the 3D track area is acquired by detecting the ground and removing points that are either too high or too low. The proposed technique was evaluated using a variety of scenes, including tunnels, elevated sections, and level urban rail transit lines. The results showed this method could successfully extract track regions from LiDAR data over significantly longer distances than conventional algorithms.

## 1. Introduction

The intrusion of obstacles into urban rail track areas is an important issue affecting transit safety. For example, on 25 December 2018, a train struck and killed a pedestrian who entered the track without permission in Shanghai Metro Line 3 [1]. On 18 March 2019, two MTR trains collided, injuring two drivers and causing serious damage to the trains [2]. On 2 April 2021, a train crashed into a construction truck that had rolled down a slope and onto the Taiwan Railway, causing several casualties [3]. On 7 January 2023, 1 person was killed and another 57 were injured in a rear-end collision in a subway in Mexico City [4]. The speed of these trains and the required braking distances often prevent any intervention by the driver. In fact, the Railway applications—Automated urban guided transport (AUGT)—Safety requirements (IEC62267) document suggests that fully automated trains with grades of automation—level 4 (GoA4) should be operated entirely without drivers [5]. As such, the development of on-board active obstacle detection technology is critical to the safe operation of urban transit systems. The primary sensor types used in on-board active obstacle detection technology include vehicle-to-vehicle communication [6,7], vision [8], and LiDAR [9,10]. Among them, vehicle-to-vehicle communication is mostly used to avoid collisions between the trains themselves, offering a long detection range, accurate distance estimations, and reduced sensitivity to external environmental factors. However, this approach is unable to detect obstacles that are not in communication with the trains.

Trains operate along fixed rails and can only apply braking measures when objects are detected in their path. Thus, track area extraction is of significance before obstacle detection. To avoid accidents and minimize the speed of potential collisions, obstacle detection technology should provide the longest measurement range possible and estimate distances to obstructions. In recent years, rapidly evolving deep learning technology and other computer vision methods have performed well in detection and segmentation tasks. However, they do exhibit certain disadvantages for distance estimation and adapting to changes in illumination, detecting only pre-trained object types. The existing detection technology, typically based on LiDAR data, encounters the same complications. Due to limitations in scanning frequency and accuracy, collected LiDAR point clouds become sparse at long distances, making it difficult to determine whether a potential obstacle is intruding into the track area or located at a safe distance. However, considering that LiDAR is less affected by light and environment, this article will extract the track area based on LiDAR.

Multi-modal data from various sensor types have been used to identify track areas in previous studies. For example, Zhang et al. [11] projected point clouds onto images by calibrating a camera with LiDAR measurements, extracting track regions using a vision algorithm. This approach relied on computer vision to provide an accurate segmentation of the tracks, which is highly sensitive to lighting and weather conditions. Guo [12] established a track map in advance using historical GPS data and applied real-time GPS information to match train locations and identify track areas. However, this technique requires real-time GPS information, which can be difficult to acquire in urban environments with multiple tunnels and buildings. There is relatively little research on the extraction of tracks using LiDAR to detect obstacles. Shen et al. [9] utilized LiDAR data but did not detect track areas, instead filtering point clouds based on the typical reflectivity of objects in rail transit lines. However, this approach removes most background points, making it difficult to eliminate interference from trackside markers. It is also feasible to identify tracks by directly detecting the rails. For example, Dai et al. [13] proposed a rule-based track plane segmentation algorithm and a region-growing background point cloud segmentation algorithm to separate the track area and the surroundings. Geng et al. [14] compared LiDAR point clouds in a railway environment, noting that features such as roughness, anisotropy, and curvature could be used to separate rail points from surrounding objects. By considering the vertical height difference between the rail surface and the ground, Zeng et al. [15] and Chen [16] identified potential rail points by detecting altitude variations in small neighborhoods. Guo et al. [17] detected rails by noting that outer grounds near the track constitute missed points, due to laser beams being blocked by the rails. Ren et al. [18] proposed a detection algorithm for low-density point clouds that used relative height, orientation, and rail gauge to roughly identify tracks through a series of filtering steps. However, these techniques are only effective over short ranges where rail points exist, which is problematic given the speed and braking distance of urban rail trains. As such, this paper aims to increase the extraction distance for the track area.

Figure 1 shows a typical LiDAR point cloud at ~150 m, where objects with good reflectivity and optimal reflection angles are clearly visible. However, it is difficult to determine whether these obstacles are inside the track area. In fact, the appearance of objects such as trackside markers and transponders are similar to that of intrusions in the point cloud. Thus, if the track area cannot be determined to exclude these items through positional relationships, it may cause false alarms. In other words, the distance over which the track area can be extracted determines the effective detection range for obstacles. In urban rail transit lines, structures parallel to the rails (e.g., tunnel walls, protective walls, and sound barriers) provide feasible references for increasing the extraction distance of track regions.

However, the existing related research faces numerous challenges in directly extracting track areas from rail point cloud data. Firstly, the rails themselves have simple geometric features and small surface areas, often appearing as sparse point sets in LiDAR scans, particularly difficult to identify at long distances. Secondly, the reflective properties of rail surfaces are often unfavorable for receiving laser signals, potentially resulting in significant data loss in point clouds at close range. Furthermore, in complex urban rail environments, adverse weather conditions (such as snow accumulation or heavy rain) and non-specific obstacles (like toolboxes or fallen rocks) can also interfere with the acquisition and processing of rail point clouds. These combined factors make it challenging for direct track area extraction methods based on rail point cloud data to meet the safety and real-time requirements in practical applications.

Urban rail transit systems commonly feature large structures parallel to the tracks, such as tunnel walls, protective barriers, and sound barriers. These structures typically have larger surface areas and more favorable reflection angles, generating more stable and dense point cloud data in LiDAR scans. Moreover, these reference objects often have good continuity and can be reliably identified at greater distances. Additionally, the radar point cloud data reflected from these fixed structures can, to some extent, mitigate interference from dynamic obstacles. Figure 2a and Figure 2b, respectively, show the three-dimensional point cloud collected by the Livox Tele-15 LiDAR in an elevated section and its projection onto a horizontal plane. As seen in Figure 3, rails can be recognized within approximately 80 m, but data points for the rails are almost completely lost beyond 100 m. In contrast, the sound barriers serving as reference objects still have a considerable number of data points at distances up to 150 m and appear as distinct curves in the two-dimensional point cloud. When the rails can no longer be detected, the reference objects can still be well identified.

In light of these observations, this study proposes a method for track area detection based on the identification and analysis of point clouds from parallel reference objects adjacent to the tracks. The method first extracts the point cloud data of parallel reference objects such as tunnel walls, protective barriers, and sound barriers using a fixed-length segmentation approach. It then selects characteristic points from these reference objects for curve fitting, generating a reference curve on either side of the track. A centerline is then derived from these two reference curves. Based on this centerline and prior knowledge, a 2D track area is expanded. Finally, through ground detection and the removal of unrealistic points, a three-dimensional track area is obtained. The effectiveness, reliability, and real-time performance of this method are validated across various point cloud data scenarios, including tunnels, elevated sections, and level ground tracks.

## 2. Materials and Methods

Figure 3 demonstrates the main process used for track area extraction proposed in this paper. First, the points formed by the references are selected in segments along the front of the train. Next, the reference points are projected onto the horizontal plane and curves are fitted on both sides. Then, after selecting a better curve from the two curves on both sides, the horizontal range of the track area is determined from the positional relationship shown in Figure 4. The point cloud within this horizontal range defines the 2D track area. Finally, the ground is fitted and points with a height that is either too high or too low are removed to obtain the 3D track area.

Figure 4 shows the positional relationships for detectable reference structures, including rails, the track centerline, and track area in the horizontal plane. These reference objects are parallel to each other. Thus, after fitting a reference curve, the track area can be obtained by translation and expansion.

The acquired point clouds contain both position information and reflection intensities for individual points. Since the LIDAR mounting position is known and its location relative to the rails is fixed, specific coordinates can be used to describe each point cloud. As shown in Figure 5, a right-handed coordinate system was established with the train forward direction forming the positive *X*-axis and the vertical upward direction serving as the positive *Z*-axis. The positive *Y*-axis was then directed to the left side of the train, with the origin located on the front face. Subsequent processing utilized these coordinates to describe the positions of each point. 

### 2.1. Reference Point Selection with Fixed-Length Segmentation

In tunnel sections, the beams emitted by the LiDAR system are completely blocked by the tunnel walls, which represent two side borders in the *Y*-axis direction of the point cloud. However, if two side borders are assumed in ground and elevated sections, points formed by pillars of the overhead line or distant tall buildings may be mistaken for reference points. Trees may also produce irrelevant points, as seen on the left side of the reference area in Figure 6. As such, reference point selection rules were established to reduce interference from ambiguous structures.

A novel technique is proposed in this study for selecting reference points based on fixed-length segmentation. This approach facilitates a more complete selection of reference points, while reducing interference from irrelevant structures such as overhead line pillars, trees, and buildings. As shown in Figure 7, point clouds were segmented using a fixed length along the *X*-axis and reference points were identified in each section. The detailed steps can be described as follows:

Step 1: Identify the point with the smallest z-coordinate in a segment of fixed length. This point forms a base and other points in the segment, whose z-coordinates are larger than a predetermined height, are discarded.

Step 2: Locate the points with the largest y-coordinates and calculate their average value to obtain a position for the left reference. Similarly, identify points with the smallest y-coordinates to establish the right reference. 

Step 3: Extend from the y-coordinates for the left and right references by a length to obtain a 2D area. All points within this region (the orange area in the figure) are selected.

Step 4: Perform the above operations on each segment to select all reference points. The selection of reference points using the above technique is demonstrated in Figure 8. The results indicate that the right reference points were selected accurately, although several irrelevant points were generated by the neighboring line.

### 2.2. Reference Curve Fitting Based on RANSAC

The sparsity of point clouds at large distances makes it difficult to represent surface information using reference points. Thus, in this paper, the reference points are projected into the horizontal plane, transforming a surface fitting problem into a curve fitting problem.

The reference points selected using the approach described in Section 2.1 were projected into the horizontal plane, thereby transforming a surface fitting problem into a curve fitting problem. Based on the shape specifications of urban rail transit lines, the coefficient of the polynomial curves was set to 2. While the least squares method is perhaps the most commonly used curve fitting algorithm, it is highly sensitive to noise [19]. Clustering was also applied during curve fitting to remove noise. However, with this approach, it can be difficult to completely retain the reference information while removing as much noise as possible. Random sample consensus (RANSAC) has been shown to identify optimal explanatory models in noisy datasets. Unlike the least squares method, RANSAC does not directly utilize all points during curve fitting. Instead, it randomly selects three points and calculates the residual of each. The curve points with residual values lying within a certain threshold are considered to be inliers. Over the course of multiple iterations, RANSAC identifies a curve containing the most inliers that best explains the point set. In this study, the residual threshold was set to 0.2 m and 100 iterations were performed on the reference points. The fitted curve is shown in Figure 9, where it is evident that RANSAC provides strong anti-noise capabilities.

### 2.3. Two-Dimensional Track Area Extraction Using Prior Knowledge of Urban Rail Transit

References on either side of the train may form points of varying quality, due to factors such as curvature, section type, and reference type. An example of this is provided in Figure 6, where reference points exhibiting poor reflection angles in the curved section are mostly absent and cannot be used. Thus, a set of rules was established to evaluate the reference curves on both sides and select a better curve from them. The 2D track area was then extracted using the process demonstrated in Figure 4.

#### 2.3.1. Better Curve Selection

The root mean square error (RMSE) measures the effect of curve fitting by calculating the difference between input points and the fitted curve. This metric can be defined as follows:(1)$$RMSE=\sqrt{\frac{1}{n}\sum_{i=1}^{n}{\left(y_{i}^{\prime }-y_{i}\right)}^{2}}$$
where n is the total number of points, $y_{i}$ is the y-coordinate of the $i^{th}$ point, and $y_{i}^{\prime }$ is the y-value of the curve for the $i^{th}$ point. In this study, RANSAC was used to fit the reference curves. We also used the least squares method to directly fit a curve to all reference points on both sides and calculated the *RMSE* to determine the proximity of these points to a curve. The number of inliers from RANSAC and the *RMSE* from the least squares method was combined to comprehensively select a better curve. This selection rule can be represented as follows:(2)$$S=\left\{\begin{array}{c} A,\left\{\begin{array}{c} f_{A}>f_{B}\times \beta \\ f_{A}\leq f_{B}\times \beta \;and\;RMSE(A)\leq RMSE(B) \end{array}\right.\\ B,\left\{\begin{array}{c} f_{B}>f_{A}\times \beta \\ f_{B}\leq f_{A}\times \beta \;and\;RMSE\left(B\right)\leq RMSE\left(A\right) \end{array}\right. \end{array}\right.$$
(3)$$f_{A}=n_{A}/N_{A}$$
(4)$$f_{B}=n_{B}/N_{B}$$
where S is the better curve, *A* and *B* are the curves of left and right references, $n_{*}$ is the number of inliers from the RANSAC reference points, $N_{*}$ is the total number of reference points, and β is a preset constant. The effects of better curve selection for different sections in Reference Point Selection with Fixed are shown in Table 1, for β = 1.2. It is evident that this rule eliminated incorrect curves, so that subsequent steps could correctly extract the track area.

#### 2.3.2. Track Centerline and Area Extraction

The origin of the point cloud coordinate system lies in the center of the track at x = 0. This track centerline can be obtained by translating the better curve along the *Y*-axis, to overlap with the origin. After acquiring the track centerline using size guidelines and relevant specifications (e.g., gauge and train width), the track area can be extracted by extending a certain length from the centerline on both sides. Taking Shanghai as an example, the “Technical Specifications of Urban Rail Transit Projects” document suggests that this extension length should be 1.3 m. Figure 10 demonstrates the effects of 2D track segmentation, where the white curve denotes the centerline. This result indicates our method has correctly identified the centerline, extracted the track area, and removed objects such as tunnel walls and trackside markers.

### 2.4. Three-Dimensional Track Area Extraction Using Ground Estimation

The 2D track areas extracted as described above do not include height information and contain objects such as rails, overhead lines, and tunnel ceilings. Since urban transit lines feature uphill and downhill sections, it is often not feasible to trim 2D track regions using a fixed height range. It is, therefore, necessary to identify the ground and the trim 2D tracks to generate 3D tracks.

By estimating the ground height and cropping point clouds over a certain vertical range, objects such as rails, overhead lines, and tunnel ceilings were removed as part of the 3D track area extraction. One common approach for estimating the ground height is to partition the point cloud using squares or sectors and then locate the ground in each partition with plane fitting. These methods typically combine uprightness, elevation, flatness, and other conditions to determine whether the fitted plane represents the physical ground. In this paper, we adopt and modify the patchwork algorithm proposed by Li [20]. The patchwork technique identifies ground sections in each partition individually. If a ground region is not found in certain partitions, irrelevant points may not be rejected in the 3D track area, which may cause false alarms. As such, we used points extracted from each partition to refit the ground region. These points were then merged using the normal direction. Under the constraint of longitudinal rail line specifications, this slope does not change frequently, and, at most, one slope change was assumed in the detection range. Thus, if no slope change occurred over this distance, all the ground points were merged into a single unit. If a slope change did occur, the ground points were merged into near and far sections. After estimating the ground location, the rail height and train height were used to define a region 0.3–3.2 m above the refitted ground plane. This space was then cropped to extract the 3D rail area. Figure 11 shows the effects of 3D extraction for the 2D rail area shown in Figure 10. It is evident that the ground, overhead lines, and tunnel ceilings were removed, as the resulting 3D track area contains only obstacles and sporadic noise points.

### 2.5. Experimental Environment

During the experimental data collection, the LiDAR sensor was mounted on a cart at a height of 1.5 m above the ground. Static and dynamic experiments were then conducted in the stations and intervals of Shanghai Metro Line 11. In the dynamic experiment, the LiDAR device moved forward with the cart at a speed of ~5 km/h. Sample photographs are shown in Figure 12, with details pertaining to each experiment provided in Table 2. Cubes with a side length of 0.5 m and staffs with a height of 1.7 m were included to mimic obstacles placed at different locations along the line. These objects exhibited high reflectivity and were easily recognizable shapes. They were placed inside or at the edges of the track area, to be used as references when verifying the correctness of the method proposed in this paper. Figure 13 shows the arrangement of obstacles for an experimental scenario, where objects in the red (green) boxes were placed inside (outside) the track area. The Livox Tele-15 LiDAR system was used in this experiment, for which the primary parameters are listed in Table 3.

## 3. Results and Discussion

As seen in the figures, the obstacles, trackside markers, and transponders formed an obvious cluster in the cloud, containing points with a higher reflection intensity. The performance of the proposed technique was evaluated by determining whether the position relationship between the extracted track area and the objects was consistent with their physical orientations. Figure 14 shows the effect of extracting track areas using the proposed technique, where the rows represent the raw point clouds, reference curves, 2D track areas, and 3D track areas, respectively. Similarly, the columns denote the outdoor straight, outdoor curved, multi-line, and tunnel sections, respectively. The extracted 3D track regions in (a) and (d) correctly retained the obstacle points, while (b) and (c) contained only sporadic noise points. Table 4 shows the results of track area extraction within 150 m, where it is apparent that the proposed method provides high accuracy in various experimental scenes. In Table 4, the accuracy for the tunnel sections was higher than that of the elevated sections, since tunnel walls constitute more points than sound barriers and protective walls. As a result, there is no interference from points outside the tunnels. By analyzing incorrect results from the outdoor scenes (i.e., elevated and ground sections), we found that recessed drains inside the track interfered with ground estimation, as shown in Figure 14. On occasion, this led to the erroneous removal of obstacles and the retention of transponders and rail segments in the extracted 3D track areas.

To comprehensively evaluate the performance of the proposed method for track area extraction based on the point clouds of trackside parallel references, we conducted a series of comparative experiments. Under the same conditions, contrast experiments were conducted between the proposed method and the traditional method. And the traditional method mainly directs the rail with LiDAR point clouds by calculating the height difference between the rails and the ground. Then, the comparative results covered multiple aspects including detection range, accuracy, processing time, environmental adaptability, and interference resistance. The experiments were carried out in various scenarios (including outdoor straight sections, outdoor curves, multi-track areas, and tunnels) and under different weather conditions (sunny, rainy, and nighttime) to thoroughly assess the effectiveness and robustness of the method.

The following Table 5 summarizes the results of the comparative experiments:

These results clearly demonstrate the significant advantages of our method across all key performance indicators, particularly in terms of detection range, accuracy, real-time performance, and environmental adaptability.

## 4. Conclusions

This paper proposed a new methodology for extracting track areas from LiDAR point clouds by detecting references commonly found in urban rail lines, specifically tunnel walls, protective walls, and sound barriers. Static experiments performed at rail stations and dynamic experiments conducted along intervals were used to evaluate the proposed technique. The results showed that this approach was able to extract 3D track areas at distances exceeding 150 m with high accuracy. In contrast, under the same experimental conditions, the rail points nearly disappeared at distances of ~100 m using conventional algorithms. Moreover, this method demonstrates certain advantages in terms of detection accuracy, real-time performance, robustness, and environmental adaptability. Its enhanced detection range, accuracy, and robustness make it particularly suitable for improving collision warning systems and for enhancing the overall operational safety of urban rail transit systems. Future work will focus on further optimizing the algorithm to address special cases and exploring its application in more diverse and challenging rail environments.

## Figures and Tables

**Figure 1 sensors-24-04963-f001:**
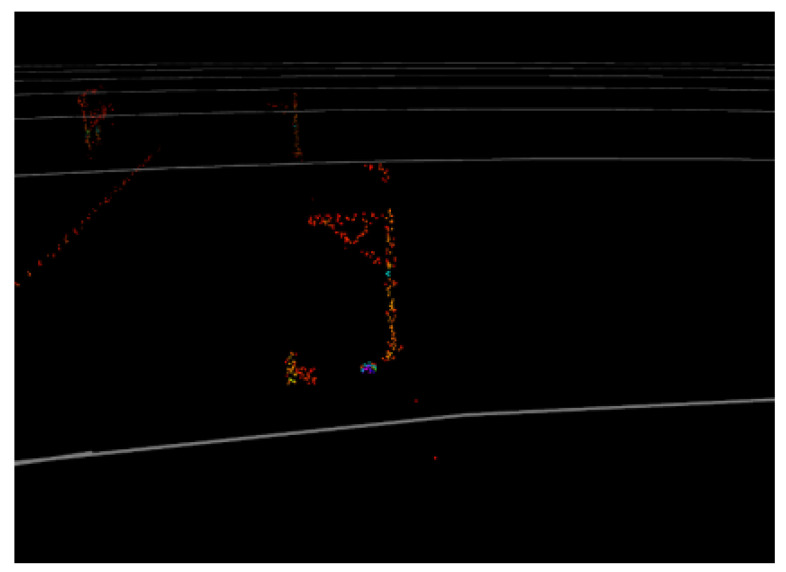
A typical LiDAR point cloud at a distance of ~150 m.

**Figure 2 sensors-24-04963-f002:**
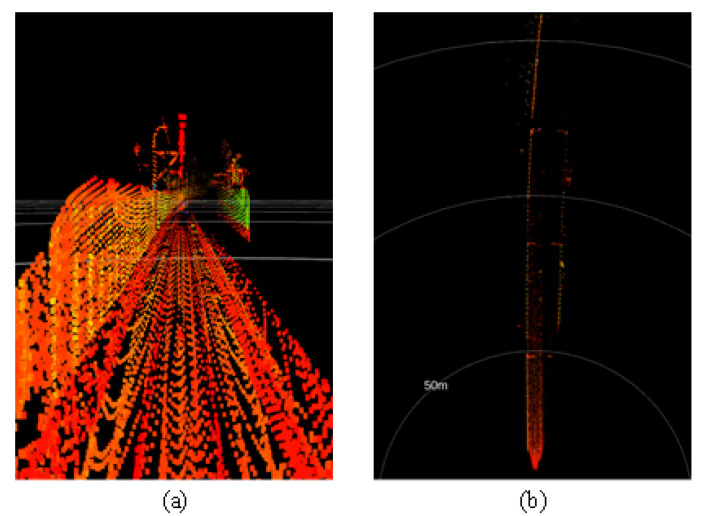
A point cloud collected by the Livox Tele-15 LiDAR system in an elevated section. (**a**) The raw point cloud. (**b**) The point cloud projected onto a horizontal plane.

**Figure 3 sensors-24-04963-f003:**
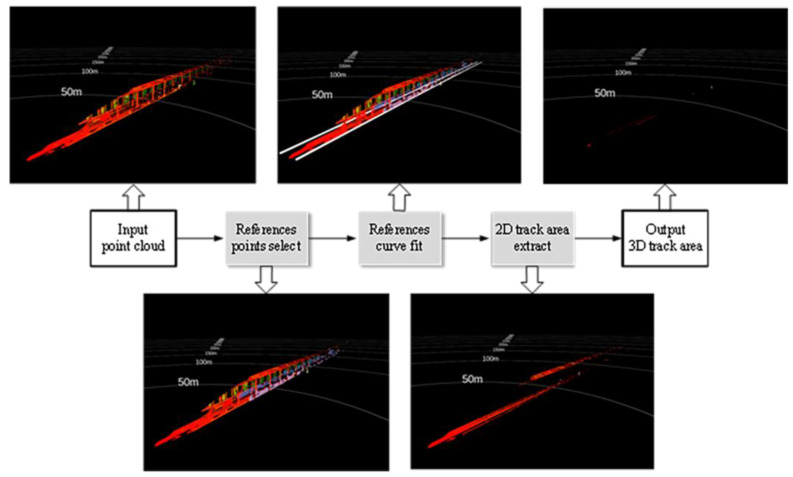
The track area extraction process.

**Figure 4 sensors-24-04963-f004:**
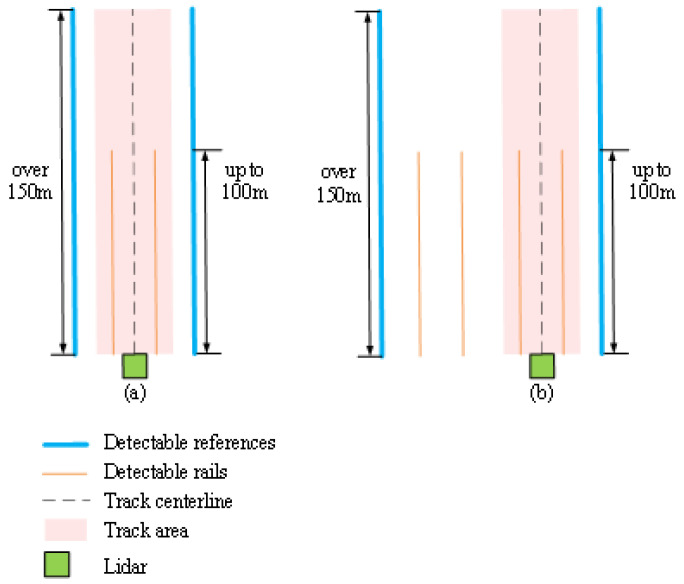
Position relationships for detectable references and rails, the track centerline, and the track area in the horizontal plane for (**a**) single-line and (**b**) double-line sections.

**Figure 5 sensors-24-04963-f005:**
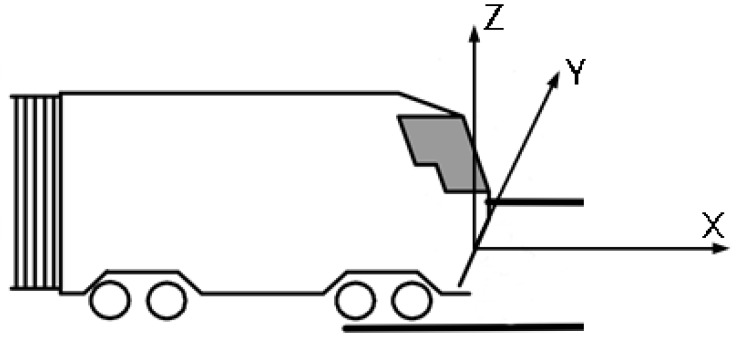
The established coordinate system. *X*-axis represents the horizontal direction of the train, the *Y*-axis indicates the vertical direction of the train, and the *Z*-axis denotes the lateral direction of the train.

**Figure 6 sensors-24-04963-f006:**
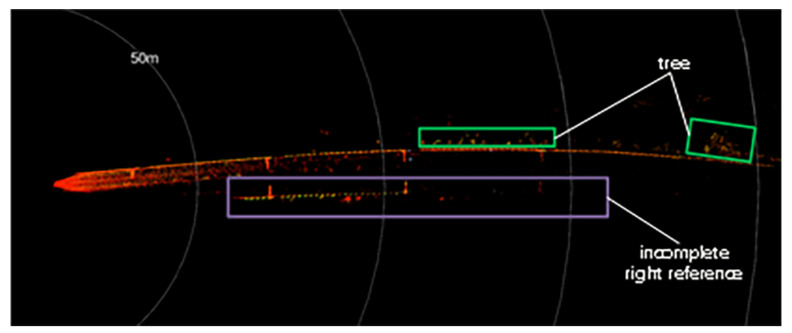
A point cloud at ground bend.

**Figure 7 sensors-24-04963-f007:**
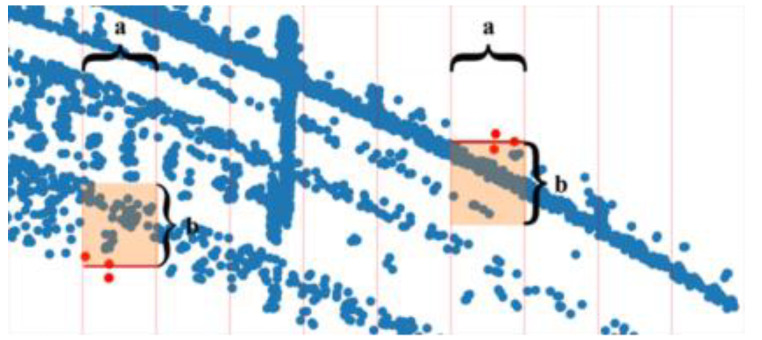
Reference points selection with fixed-length segmentation.

**Figure 8 sensors-24-04963-f008:**
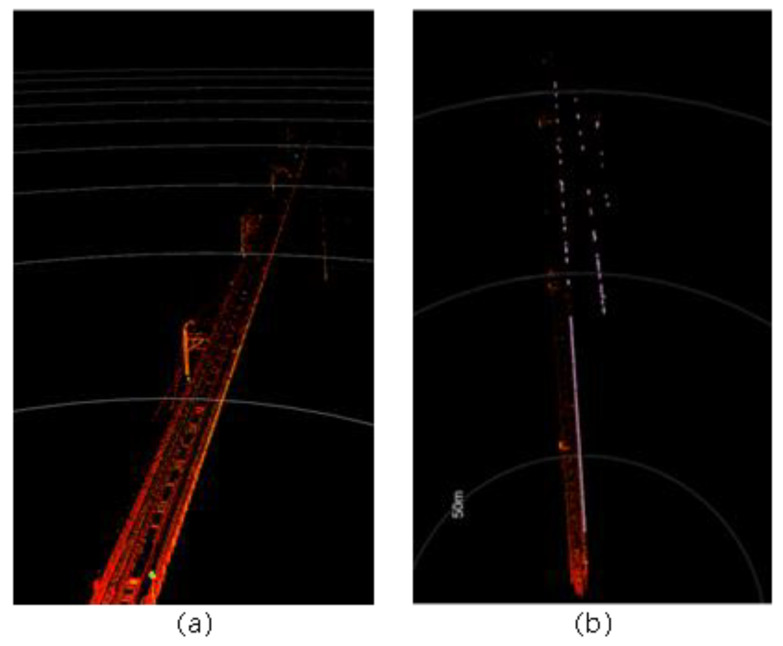
The effect of right reference point selection. The fixed length a = 1 m, the extended length b = 0.2 m, and the number of sampling points k = 3. (**a**) The raw point cloud. (**b**) Reference point cloud extraction results.

**Figure 9 sensors-24-04963-f009:**
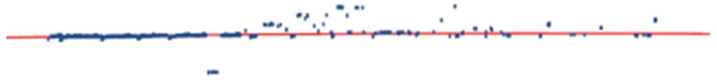
The effects of using RANSAC for curve fitting.

**Figure 10 sensors-24-04963-f010:**
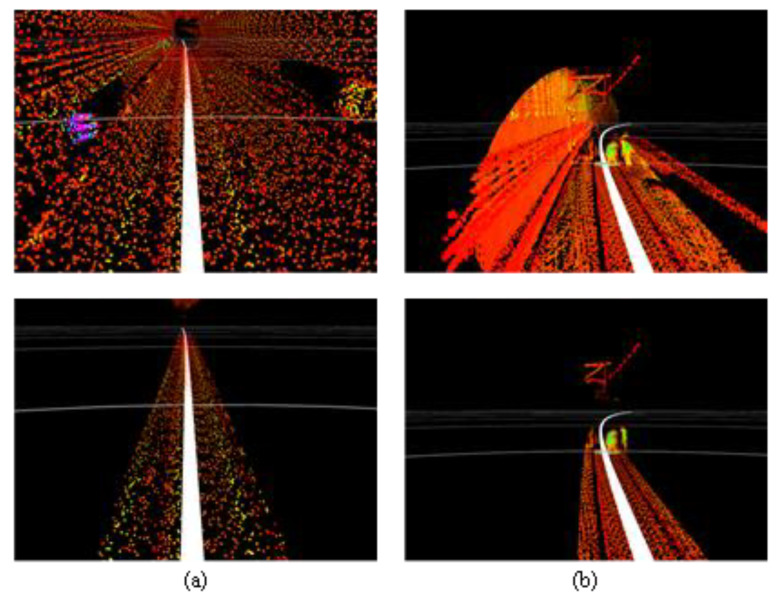
The effects of 2D track area extraction for (**a**) tunnel and (**b**) ground sections.

**Figure 11 sensors-24-04963-f011:**
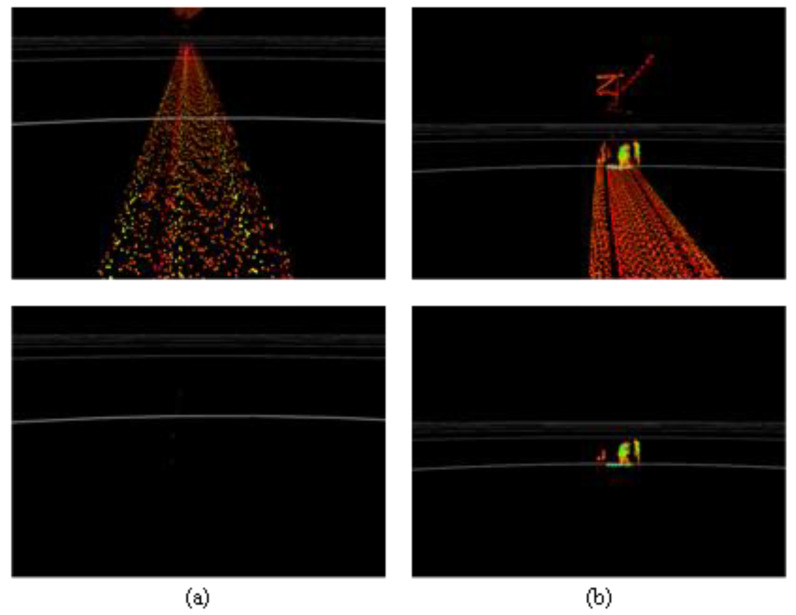
The effect of 3D track area extraction for (**a**) tunnel and (**b**) ground sections.

**Figure 12 sensors-24-04963-f012:**
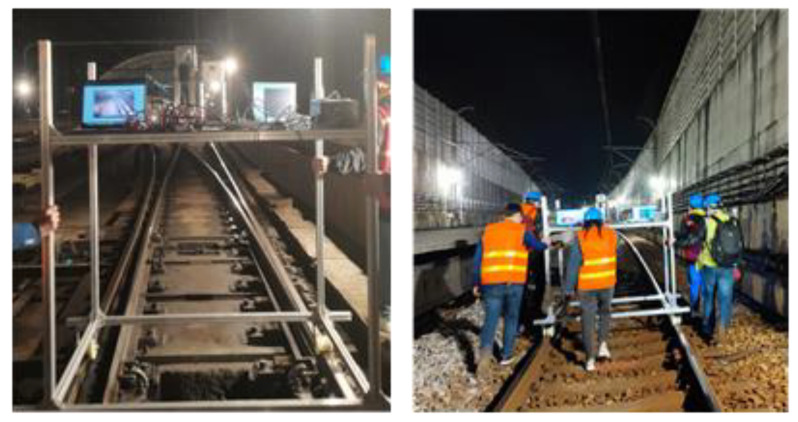
Sample experimental photographs.

**Figure 13 sensors-24-04963-f013:**
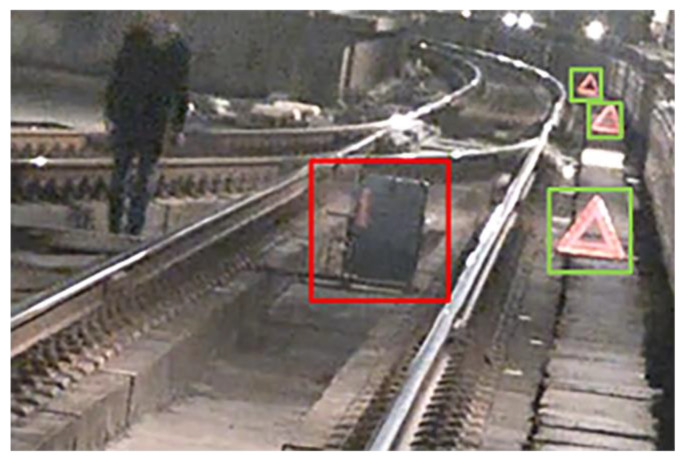
The arrangement of obstacles in an experimental setting.

**Figure 14 sensors-24-04963-f014:**
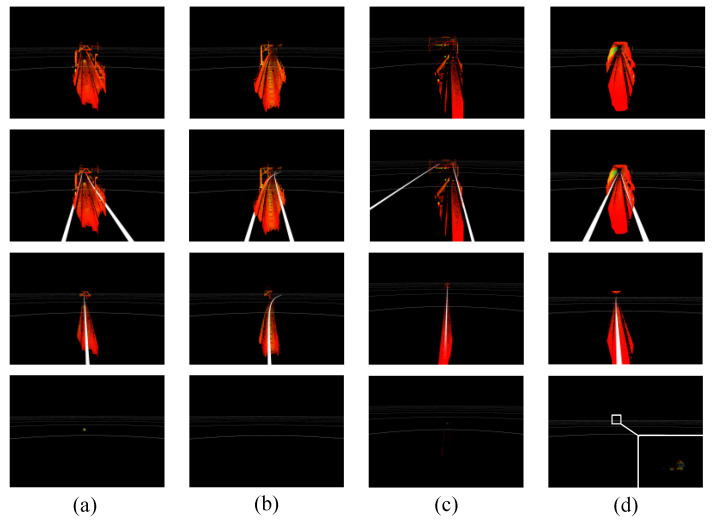
The effects of track area extraction in different sections. The extracted 3D track regions in (**a**) and (**d**) correctly retained the obstacle points, while (**b**) and (**c**) contained only sporadic noise points.

**Table 1 sensors-24-04963-t001:** The effect of better curve selection in different sections.

Point Cloud with Curves	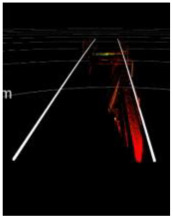		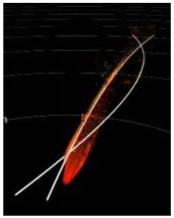		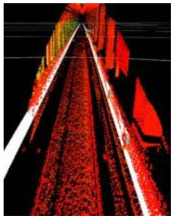	
Inliers ratio	left 0.442	right 0.810	left 0.955	right 0.478	left 0.756	right 0.676
RMSE	left 4.707	right 2.430	left 0.093	right 3.321	left 0.790	right 1.020
Better curve	Right		Left		Left	

**Table 2 sensors-24-04963-t002:** Detailed experiment locations.

Experiment Type	Experimental Spot	Section Type
Static	Station A	Elevated section
	Station B	Tunnel section
Dynamic	Station A to Station C interval	Tunnel, elevated, and ground section

**Table 3 sensors-24-04963-t003:** Livox Tele-15 LiDAR parameters.

Parameters	Value
Laser wavelength	905 nm
Field of view	14.5°×16.2°
Range precision	2 cm
Angular precision	<0.03°
Beam divergence	0.12°×0.02°
Data volume	240,000 points per second

**Table 4 sensors-24-04963-t004:** Experimental results.

Experimental Scenes		Frame	Accuracy
Static experiment	Tunnel	1059	97.3%
	Elevated	1130	93.2%
Dynamic experiment		8710	94.8%

**Table 5 sensors-24-04963-t005:** Comparative experiments results.

Performance Metric	Proposed Method	Traditional Method
Maximum Effective Detection Range	150 m (up to 200 m in tunnels)	100 m
Average Accuracy	95.2%	75.3%
Accuracy in Tunnel Sections	97.4%	80.4%
Accuracy in Elevated Sections	92.8%	71.9%
Accuracy in Ground Sections	93.7%	73.1%
Average Processing Time	50 ms/frame	120 ms/frame
Performance Degradation in Various Weather Conditions	<5%	>20%
Processing Frequency on Embedded Systems	15 Hz	6 Hz

## Data Availability

The original contributions presented in the study are included in the article; further inquiries can be directed to the corresponding authors.

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
