# Peer review of "LiDAR-Based Urban Three-Dimensional Rail Area Extraction for Improved Train Collision Warnings"

_sensors, 2024, doi:10.3390/s24154963_

Round 1

Reviewer 1 Report

Comments and Suggestions for Authors

It is obvious that the current status of obstacle detection technology in front of trains mentioned in this article is not the latest. There are already many application scenarios that use LiDAR and image recognition technology for actual subway lines, such as Beijing Metro Line 11, Hong Kong Tsuen Wan Line, etc. The 3D point cloud technology method proposed in this article does not see the specific application scenarios and specific problems to be solved. It is the problem of detecting what kind of obstacles ahead the train is running at what speed, and the impact of discovering obstacles ahead on the train's operation status. In addition, quantitative analysis has not been seen, and most of the figures in the article display point cloud images, lacking interpretation and analysis of the images.

Comments on the Quality of English Language

The English expression ability reflected in this article is still acceptable

Author Response

Dear Editor and Reviewers,

We sincerely appreciate your insightful feedback regarding the computational aspects of our proposed algorithm. We agree that this is a crucial aspect, especially given the large size of point cloud data and the need for real-time processing in collision warning systems. In response to your valuable comment, we have made the following additions and revisions to our manuscript:

  1. Point cloud data processing: We further optimize the proposed point cloud data processing methods, such as reference point extraction based on fixed-length segmentation and point cloud segmentation based on ground identification, to enhance the computational feasibility and computational efficiency of the algorithms in terms of computation, and the experimental results show that the proposed algorithms are capable of processing large-scale point cloud data of the track field.
  2. Comparative experiments: To further strengthen our work, we have conducted additional comparative experiments. These experiments compare our method with traditional direct rail detection approaches across various scenarios and environmental conditions. The results, now included in the manuscript, demonstrate the superior performance of our method in terms of detection range, accuracy, real-time capability, and robustness.
  3. Future work: We have expanded the discussion on future work, addressing the optimization of the algorithm for special cases and its potential application in more diverse and challenging rail environments.

These additions provide a comprehensive discussion of the computational aspects of our proposed algorithm, addressing the concerns raised in your feedback. We believe these changes significantly enhance the quality and practical relevance of our paper.

We appreciate your attention to this critical aspect of our work and thank you for the opportunity to improve our manuscript. We look forward to your thoughts on these additions.

Sincerely,

Tengfei Yuan

Reviewer 2 Report

Comments and Suggestions for Authors

The paper proposes a collision warning system using LiDAR data. The proposed algorithm is shown to work on real-world data. The results are promising. However, a more thorough discussion on the computational aspects of the proposed algorithm would enhance the quality of the paper. The point cloud data are huge in size while objects of interest only present part of the large data. Therefore, computational feasibility of any algorithm is of paramount importance. It would be good to see comments on this aspect.

Comments on the Quality of English Language

The language quality is adequate. 

Author Response

Dear Editor and Reviewers,

We greatly appreciate your valuable feedback on our manuscript. Your comments have played a crucial role in improving the quality and relevance of our work. We have carefully addressed each point raised and made substantial revisions to the manuscript. Below is our detailed response to your comments:

  1. Update on current obstacle detection technology: We recognize that the literature review in our initial draft did not fully reflect the latest developments in obstacle detection technology for urban rail systems. We have comprehensively updated this section, including recent applications in Beijing Metro Line 11 and Hong Kong Tsuen Wan Line. This revision provides a more comprehensive and up-to-date context for our research.
  2. Specific application scenarios and problems: We have clarified the specific application scenarios and problems addressed by our proposed method, namely the automatic extraction of track areas within a 150m range for urban rail trains operating at speeds of around 80km/h, in order to improve the accuracy of the detection of obstacles in front of the train and to reduce the false alarm rate, so as to achieve the purpose of enhancing the performance of on-board track obstacle detection algorithms.
  3. Quantitative analysis and image interpretation: In response to this valid concern, we have strengthened our quantitative analysis. We have provided detailed performance metrics, including detection accuracy, processing time, and robustness under different environmental conditions.
  4. Comparative experiments: To further strengthen our work, we have conducted additional comparative experiments. These experiments compare our method with traditional direct rail detection approaches across various scenarios and environmental conditions. The results, now included in the manuscript, demonstrate the superior performance of our method in terms of detection range, accuracy, real-time capability, and robustness.
  5. Future work: We have expanded the discussion on future work, addressing the optimization of the algorithm for special cases and its potential application in more diverse and challenging rail environments.

These revisions have substantially enhanced the manuscript, providing a more comprehensive, up-to-date, and analytically robust presentation of our research. We believe these changes will significantly improve the quality and relevance of our paper.

We thank you again for your insightful comments and the opportunity to improve our work. We look forward to your feedback on these revisions.

Sincerely,

Tengfei Yuan

Round 2

Reviewer 1 Report

Comments and Suggestions for Authors

In page 12 table 5, what does the traditional method refer to in the table, and what do these indicators of accuracy refer to need to be clear.

Comments on the Quality of English Language

Some sentences should be improved. Such as page 13 line 353,Results showed this approach was able to extract 3D track ares at distances exceeding 150 meters with high accuracy.

Author Response

Thanks for your comments. The specific responses are as follows:

Under the same conditions, the contrast experiments are conducted between the proposed method and the traditional method. And the traditional method mainly direct the rail with LiDAR point clouds by calculating the height difference between the rails and the ground. Then, the comparative results cover multiple aspects including detection range, accuracy, processing time, environmental adaptability, and interference resistance.